# Differences in Sources of Information, Risk Perception, and Cognitive Appraisals between People with Various Latent Classes of Motivation to Get Vaccinated against COVID-19 and Previous Seasonal Influenza Vaccination: Facebook Survey Study with Latent Profile Analysis in Taiwan

**DOI:** 10.3390/vaccines9101203

**Published:** 2021-10-19

**Authors:** Yi-Lung Chen, Yen-Ju Lin, Yu-Ping Chang, Wen-Jiun Chou, Cheng-Fang Yen

**Affiliations:** 1Department of Healthcare Administration, Asia University, Taichung 41354, Taiwan; elong@asia.edu.tw; 2Department of Psychology, Asia University, Taichung 41354, Taiwan; 3Department of Psychiatry, School of Medicine, College of Medicine, Kaohsiung Medical University, Kaohsiung 80708, Taiwan; 1040457@kmuh.org.tw; 4Department of Psychiatry, Kaohsiung Medical University Hospital, Kaohsiung 80708, Taiwan; 5School of Nursing, The State University of New York, University at Buffalo, New York, NY 14214-8013, USA; yc73@buffalo.edu; 6School of Medicine, Chang Gung University, Taoyuan 33302, Taiwan; 7Department of Child and Adolescent Psychiatry, Chang Gung Memorial Hospital, Kaohsiung Medical Center, Kaohsiung 83301, Taiwan; 8College of Professional Studies, National Pingtung University of Science and Technology, Pingtung 91201, Taiwan

**Keywords:** COVID-19, vaccine, motivation, seasonal influenza

## Abstract

The present study aimed (1) to identify distinct latent classes of motivation to get vaccinated against coronavirus disease 2019 (COVID-19) and previous seasonal influenza vaccination among people in Taiwan and (2) to examine the roles of sources of information, risk perception, and cognitive appraisals of vaccination against COVID-19 in these classes. We recruited 1047 participants through a Facebook advertisement. The participants’ motivation to get vaccinated against COVID-19, previous seasonal influenza vaccination, sources of information about COVID-19 vaccination, risk perception of COVID-19, and cognitive appraisals of vaccination against COVID-19 were determined. We examined the participants’ motivation for COVID-19 vaccination and previous seasonal influenza vaccination through latent profile analysis. Four latent classes of motivation were identified: participants with high motivation for COVID-19 vaccination and high seasonal influenza vaccination, those with high motivation for COVID-19 vaccination but low seasonal influenza vaccination, those with low motivation for COVID-19 vaccination but high seasonal influenza vaccination, and those with low motivation for COVID-19 vaccination and low seasonal influenza vaccination. Compared with participants in the latent class of high motivation for COVID-19 vaccination and high seasonal influenza vaccination, those in the other three latent classes had lower levels of positive appraisals of COVID-19 vaccination; participants in the latent class of low motivation for COVID-19 vaccination and low seasonal influenza vaccination had lower risk perception of COVID-19 and were also less likely to obtain information about COVID-19 vaccination from the internet, friends, and family members. The various motivations and behaviors for vaccination, sources of information, risk perception, and cognitive appraisals of vaccination against COVID-19 should be considered in intervention programs aiming to increase people’s motivation to get vaccinated against COVID-19.

## 1. Introduction

### 1.1. Motivation to Get Vaccinated against Coronavirus Disease 2019 (COVID-19)

Coronavirus disease 2019 (COVID-19) has had a disastrous effect worldwide [1]. The vaccines against COVID-19 are expected to reduce the spread, severity, and death caused by COVID-19 [2,3]. However, in addition to the unequal distribution of vaccines worldwide [4,5] and uncertain effects of the vaccines on new COVID-19 variants [6], vaccine hesitancy is an imminent challenge [7,8,9,10]. People with vaccine hesitancy delay accepting or refuse vaccines despite the availability of vaccination services [11]. Multiple factors relate to the motivation to vaccinate. By using the Health Belief Model (HBM) [12], previous studies have demonstrated that the risk perception of COVID-19 and perceived benefits and barriers of vaccination for COVID-19 were significantly associated with the motivation to receive COVID-19 vaccination [8,13,14,15,16]. By using the Theory of Planned Behavior [17], previous studies have found the associations of attitude, subjective norms, and perceived behavioral control with the motivation to get vaccinated against COVID-19 [13,18]. By using the Protection Motivation Theory [19], perceived knowledge was significantly associated with intention to get vaccinated against COVID-19 [20]. In addition, the sources from which individuals obtain information on vaccination for infectious diseases play a crucial role in their attitudes toward vaccination [21,22,23]. Research also revealed that critical social and international events, as well as the announcements from political leaders and authorities, may influence the attitude toward vaccination in the public [24,25]. Especially, people may be exposed to conspiracy theories and emotional viewpoints about vaccination against COVID-19 from social media [25]. The predictors of the motivation to get vaccinated against COVID-19 should be identified to develop effective strategies for enhancing the public’s motivation to get vaccinated against COVID-19.

### 1.2. Previous Seasonal Influenza Vaccination and the Motivation to Get Vaccinated against COVID-19

The role of previous seasonal influenza vaccination in the prediction of the motivation to vaccinate against COVID-19 warrants further study. Several studies have demonstrated that previous seasonal influenza vaccination may reduce the morbidity and mortality in patients with COVID-19 [26,27,28,29]. Seasonal influenza vaccination before the COVID-19 pandemic indicates a specific behavior that demonstrates practical acceptance of the effectiveness and safety of vaccines. A previous study in Taiwan revealed that the risk perception of influenza infection was positively associated with intention to receive a vaccine against influenza [30]. Moreover, research has demonstrated that previous vaccination against seasonal influenza before the COVID-19 pandemic positively predicted the motivation to get vaccinated against COVID-19 in the general population [31,32,33,34] and among health care workers [35,36], thus supporting that the individuals without vaccination against seasonal influenza should be the target of intervention programs for enhancing their motivation to get vaccinated against COVID-19.

Despite significant correlations between previous seasonal influenza vaccination and the motivation to get vaccinated against COVID-19, a proportion of individuals showed motivation for COVID-19 vaccination that did not correspond with their experiences of previous vaccination against seasonal influenza [26,27,28,29]. This noncorrespondence may be due to the different development processes of vaccines for seasonal influenza and COVID-19 and temporal and spatial differences. Identifying the distinct classes of the motivation to get vaccinated against COVID-19 and previous seasonal influenza vaccination, and examining the factors related to the various classes of motivation, may provide knowledge for developing prevention strategies against COVID-19.

### 1.3. Study Aims and Hypotheses

The present study aimed (1) to identify the distinct latent classes of the motivation to get vaccinated against COVID-19 and previous seasonal influenza vaccination among people in Taiwan and (2) to examine the roles of information sources, risk perception, and cognitive appraisals of vaccination against COVID-19 across these classes. We proposed two hypotheses. First, we hypothesized that individuals can be categorized into various latent classes according to their motivation to get vaccinated against COVID-19 and previous seasonal influenza vaccination. Second, we hypothesized that sources of information, risk perception, and cognitive appraisals of vaccination against COVID-19 differ among the classes. Research has found that being female [9], of a younger age [37,38], and of lower educational attainment [37] were significantly associated with lower motivation to receive COVID-19 vaccination; therefore, we adjusted the effects of gender, age, and education level when examining the differences in sources of information, risk perception, and cognitive appraisals between various latent classes.

## 2. Methods

### 2.1. Participants

The procedure for recruiting participants in this study has been described elsewhere [39]. In brief, 1047 participants were recruited by using a Facebook advertisement on 15 October 2020 and 21 December 2020. Facebook (Facebook Inc., Menlo Park, CA, USA) users were eligible for this study if they were ≥20 years old and living in Taiwan. The Facebook advertisement (see Appendix A) included a headline, main text, pop-up banner, and weblink to the research questionnaire website. We designed the advertisement to appear in the “News Feed” of Facebook, which is a streaming list of updates from the user’s connections (e.g., friends) and advertisers. We focused solely on News Feed advertisements (Facebook) as opposed to other Facebook advertising locations (e.g., the right column) because News Feed advertisements are more effective in terms of recruitment metrics for research studies. We targeted the advertisement to Facebook users by location (Taiwan) and by language (Mandarin Chinese) such that a given advertisement appeared on a user’s news feed as determined by a Facebook algorithm. As of 21 December 2020, in Taiwan, 627 patients had had COVID-19, and 7 patients had died of COVID-19 [40]. No vaccine against COVID-19 was available in Taiwan during the study period. The Institutional Review Board of Kaohsiung Medical University Hospital approved this study (KMUHIRB-EXEMPT(I) 20200019).

### 2.2. Measures

#### 2.2.1. Motivation to Get Vaccinated against COVID-19

The motivation to get vaccinated against COVID-19 was assessed using one item. The question and scoring are presented in Table 1. A higher score indicated a higher motivation to get vaccinated against COVID-19 [41].

#### 2.2.2. Previous Seasonal Influenza Vaccination

The previous vaccination for seasonal influenza was assessed using one question: “Did you receive seasonal influenza vaccination in the recent years before the COVID-19 pandemic?”, with the scores ranging from 1 (never) to 4 (always).

#### 2.2.3. Sources of Information about COVID-19 Vaccine

We asked participants whether they had obtained COVID-19 vaccination information from internet media (e.g., Facebook, Twitter, blogs, and internet news), traditional media (e.g., newspapers, television, and radio broadcasting), friends, and family members [41]. The questions and scoring are listed in Table 1. Participants were divided into those who had never, or those who had ever, obtained COVID-19 vaccination information from that source.

#### 2.2.4. Risk Perception

A five-item questionnaire was used to measure the perception of risk regarding contracting COVID-19 [42]. The five items assessed the following: worry if developing flu-like symptoms, worry about the possibility of contracting COVID-19, worry about COVID-19, perceived likelihood of contracting COVID-19, and perceived chance of contracting COVID-19 compared with others outside their family. The questions and scoring are provided in Table 1. A higher total score indicated greater risk perception. Cronbach’s α was 0.704 in this study.

#### 2.2.5. Drivers of COVID-19 Vaccination Acceptance Scale

The Drivers of COVID-19 Vaccination Acceptance Scale [43] was adapted from the Motors of Influenza Vaccination Acceptance Scale [44], and this scale measures the four cognitive appraisals of vaccination against COVID-19: values (three items, i.e., caring about the purpose of COVID-19 vaccination), impacts (three items, i.e., belief in the effects of COVID-19 vaccination uptake in preventing COVID-19 infection), knowledge (three items, i.e., knowledge regarding COVID-19 vaccination), and autonomy (three items, i.e., confidence in and control over getting a COVID-19 vaccination if desired). The questions and scoring are listed in Table 1. Cronbach’s α of the four domains ranged from 0.682 to 0.896 in this study.

To reduce bias that may have been introduced by the data collection from the Facebook capture method, and to justify the internal consistency and reliability of the survey data from the Facebook capture method [45], we conducted a Cronbach’s α for all questionnaire items used in this study. Cronbach’s α of the all questionnaire items was 0.711, indicating reasonably high reliability of these data.

#### 2.2.6. Sociodemographic Characteristics

Data on sex, age (<35, 35–49, and ≥50 years), educational level (high school or below, bachelor’s degree, and master’s degree and above), and occupation (health care workers and non-health care workers) were collected.

### 2.3. Statistical Analysis

We explored participants’ motivation to get vaccinated against COVID-19 and previous seasonal influenza vaccination through latent profile analysis (LPA) by using the R package *tidyLPA* [46]. The LPA focuses on identifying a subsystem by measuring its components and classifying individuals into several subgroups based on indicator variables. Two indicator variables were used in this study: (1) the motivation to get vaccinated against COVID-19 and (2) previous seasonal influenza vaccination. To avoid the scale difference between variables and to increase the comparability between variables, we first standardized two indicator variables to make their values with zero-mean and unit variance. To determine how many latent classes obtained from LPA, we used the basic model according to four model fit indices: Akaike information criterion (AIC), Bayesian information criterion (BIC), entropy, and bootstrapped likelihood ratio test (BLRT). The model with lower AIC and BIC has a better fit than those with higher AIC and BIC. An entropy value approaching 1 indicates a clear separation of classes [47], and entropy >0.80 indicates that the latent classes are highly discriminating [48]. For BLRT, *p* < 0.05 indicates that the *k* class model is superior to the *k* − 1 class model (*k* represents the number of classes). If there was no best fit model, we would select the number of latent classes based on the parsimony principle and the literature.

The differences in sociodemographics (including sex, age, educational level, and occupation) between various latent classes of the motivation to get vaccinated against COVID-19 and previous seasonal influenza vaccination were examined using multiple multinomial logistic regression with the latent class as the nominal outcome variable. The differences in sources of information about COVID-19 vaccination, perception of COVID-19 risk, and cognitive appraisals of vaccination against COVID-19 between the latent classes were examined using multiple multinomial logistic regression with adjustment for sociodemographics.

## 3. Results

### 3.1. Results of LPA

Table 2 presents the results of the model fit indices in the LPA. No model with the best fit could be consistently found based on minimizing the AIC and BIC and other model indexes in the LPA. Although the model with two latent classes had an inflection point of the BIC, the BIC and AIC continuously decreased when the number of classes increased; the statistical significance was based on the BLRT. It is not rational to select an LPA model with a great number of classes, especially for a model that has only two variables (i.e., motivation to get vaccinated against COVID-19 and previous seasonal influenza vaccination). Thus, we chose the best model based on the parsimony principle and the domain knowledge of the literature [49]; the number of classes in an LPA usually ranges from two to four. The three-class model was not considered because of its poorer performance based on entropy. Finally, we selected the four-class model because it could more effectively discern the difference in motivation to get vaccinated against COVID-19 and previous seasonal influenza vaccination between the classes and given its adequate performance indicated by the model fit indexes.

Figure 1 presents the standard scores of the motivation to get vaccinated against COVID-19 and previous seasonal influenza vaccination per latent class. The first latent class (35.0% of the sample, 366/1047) was named “high motivation for COVID-19 vaccination and high seasonal influenza vaccination” (the latent class of Both High) and comprised participants with high scores for the motivation to get vaccinated against COVID-19 and previous seasonal influenza vaccination. The second latent class (33.8% of the sample, 354/1047) was named “high motivation for COVID-19 vaccination but low seasonal influenza vaccination” (the latent class of High COVID-19 but Low Influenza) and consisted of those with high scores for the motivation for vaccination against COVID-19 but low scores for vaccination against seasonal influenza. The third latent class (12.8% of the sample, 134/1047) was named “low motivation for COVID-19 vaccination but high seasonal influenza vaccination” (the latent class of Low COVID-19 but High Influenza) and consisted of those with low scores for the motivation for vaccination against COVID-19 but high scores for seasonal influenza vaccination. The fourth latent class (18.4% of the sample, 193/1047) was named “low motivation for COVID-19 vaccination and low seasonal influenza vaccination” (the latent class of Both Low) and consisted of those with low scores for the motivation for vaccination against COVID-19 and seasonal influenza vaccination.

### 3.2. Variables Predicting Latent Classes

Table 3 presents the results of the multiple multinomial logistic regression analysis of the differences in sociodemographics, sources of information, risk perception, and cognitive appraisals of vaccination against COVID-19 between the latent classes. Based on the profiles of the latent classes from the LPA, we selected the class of “high motivation for COVID-19 vaccination and high seasonal influenza vaccination” as the reference group in the multiple multinomial logistic regression analysis because it may be more crucial to identify individuals who had low motivation of vaccination for COVID-19 and seasonal influenza for the public interest. Firstly, differences in sex, age, educational level, and occupation between various latent classes were examined using unadjusted multinomial logistic regression. Compared with participants in the latent class of Both High, those in the latent class of High COVID-19 but Low Influenza were more likely to belong to the age groups of <30 or 35–49 years, possess an educational level of high school or below, and be non-health care workers; those in the latent class of Low COVID-19 but High Influenza were more likely to be female; and those in the latent class of Both Low were more likely to have the educational level of high school or below and be non-health care workers.

The differences in the information sources between various latent classes were further examined using adjusted multinomial logistic regression. Because of significant differences in sex, age, educational level, and occupation between the latent classes, these sociodemographics were adjusted in multinomial logistic regression. Compared with participants in the latent class of Both High, those in the other three latent classes had lower levels of impact, knowledge, value, and autonomy of vaccination against COVID-19; however, the ORs for the class of High COVID-19 but Low Influenza (0.89 to 0.95) were greater than those for the class of Low COVID-19 but High Influenza (0.64 to 0.80) and the class of Both Low (0.60 to 0.80). Moreover, the participants in the latent class of Both Low had lower risk perception of COVID-19; they were also less likely to obtain information about COVID-19 vaccination from the internet, friends, family members, or traditional media.

## 4. Discussion

The present study identified four latent classes of the motivation to get vaccinated against COVID-19 and previous seasonal influenza vaccination: Both High, High COVID-19 but Low Influenza, Low COVID-19 but High Influenza, and Both Low. Several sources of information about COVID-19 vaccination, risk perception of COVID-19, and cognitive appraisals of vaccination against COVID-19 differed between these latent classes.

### 4.1. Latent Classes of Both High and Both Low

The present study identified the latent classes of Both High and Both Low, in which the participants had corresponding levels of motivation to get vaccinated against COVID-19 and previous seasonal influenza vaccination, respectively. These two latent classes are in line with the results of prior studies that demonstrated previous seasonal influenza vaccination predicts motivation to get vaccinated against COVID-19 during the pandemic [31,32,33,34]. Individuals in the latent class of Both Low may have the highest risk of contracting COVID-19 and seasonal influenza. Previous seasonal influenza vaccination could reduce the morbidity and mortality in patients with COVID-19 [26,27,28,29]; therefore, it is important to enhance the motivation to get vaccinated against COVID-19 and seasonal influenza. Compared with participants in the latent class of Both High, those in the latent class of Both Low had lower perceptions of COVID-19 risk, values, and positive impacts of COVID-19 vaccination, and their knowledge about and autonomy regarding vaccination against COVID-19 were also lower. The results indicated that governments and health professionals should actively promote individuals’ cognitive appraisals of COVID-19 vaccination. However, the present study found that, compared with those in the latent class of Both High, those in the class of Both Low were less likely to obtain information of COVID-19 vaccination from the internet, friends, or family members. How to effectively deliver intervention programs for promoting COVID-19 vaccination among individuals with low motivation for COVID-19 vaccination and low seasonal influenza vaccination warrants careful planning.

### 4.2. Latent Classes of High COVID-19 but Low Influenza and Low COVID-19 but High Influenza

The present study identified two latent classes of participants with noncorresponding levels of the motivation to get vaccinated against COVID-19 and previous seasonal influenza vaccination, named High COVID-19 but Low Influenza and Low COVID-19 but High Influenza. The results support that previous vaccination against seasonal influenza does not guarantee a high motivation to get vaccinated against COVID-19. As the most severe respiratory infectious disease (RID) since the development and use of vaccines, COVID-19 has changed people’s attitudes toward preventive behaviors against RIDs globally [50]. The results of the present study revealed that individuals in the class of High COVID-19 but Low Influenza might shift their attitude to favoring vaccination after experiencing the COVID-19 pandemic. Several studies have revealed that the COVID-19 pandemic increased the intention to get vaccinated against seasonal influenza [50,51,52]. Notably, in this study, participants in the class of High COVID-19 but Low Influenza had lower cognitive appraisals of COVID-19 vaccination than did those in the class of Both High, indicating basic differences in the beliefs related to vaccination between these two groups.

The vaccines for seasonal influenza have been developed and are used to prevent the spread of seasonal influenza, with proven benefits. In Taiwan, seasonal influenza mass vaccination campaigns have been conducted annually in the past two decades [53,54]. By contrast, the COVID-19 vaccines have been developed in a short period, causing some individuals to be skeptical. Moreover, some complications of the vaccines against COVID-19 have been reported [55]. Whether the development processes of vaccines and people’s misgiving toward the complications of vaccines account for the hesitant attitude toward COVID-19 vaccination in the class of Low COVID-19 but High Influenza warrants further study. Especially, health care workers were the main members (53.7%) of the latent class of Low COVID-19 but High Influenza. Whether the hesitancy to COVID-19 vaccination in health care workers may negatively influence the prevention of COVID-19 needs to be monitored closely.

### 4.3. Implications

The present study is the first one to classify individuals with various levels of motivation to get vaccinated against COVID-19 and previous seasonal influenza vaccination and to examine the differences in information sources, risk perception, and cognitive appraisal of COVID-19 vaccination between the various classes of motivation. The results can be referenced to develop intervention programs to subtly enhance the motivation to get vaccinated against COVID-19 among people from various groups. First, governments and health professionals should actively promote the cognitive appraisals of COVID-19 vaccination in the individuals with low motivation for COVID-19 vaccination and low seasonal influenza vaccination. As the cognitive appraisal contains values, impacts, knowledge, and autonomy about vaccination against COVID-19, the intervention programs for enhancing people’s cognitive appraisal should integrate multiple components to increase the effectiveness. Moreover, the intervention programs should be delivered via multiple channels and not via a sole channel, such as the internet or advertisements via traditional media.

### 4.4. Limitations

The present study had some limitations. First, delivering study questionnaires through social media can provide a large number of responses in a short period [56], particularly during the pandemic. However, Facebook has a younger audience than the general population [56,57]. The participants who were 50 years old or older accounted for only 9.2% of all the participants in the present study. Older adults are the main target for both COVID-19 and seasonal influenza vaccines; they might be underrepresented due to the recruitment on Facebook. Moreover, social media users are also more exposed to incorrect information on vaccines [25]. Therefore, the results in our study may not be generalizable to the general population. Second, the motivation to get vaccinated against RIDs can be easily influenced by the severity of the pandemic. Longitudinal studies are needed to monitor the changes in people’s motivations and behaviors to actually get vaccinated. Third, research found that medical professionals are the most reliable source of health-related information regarding COVID-19 [58]. However, given that 26.6% of the participants in this study were health care workers, we did not analyze the role of getting information of COVID-19 vaccines from medical professionals in the latent classes of motivation to get vaccinated against COVID-19 and previous seasonal influenza vaccination.

## 5. Conclusions

The present study identified four latent classes of motivations to get vaccinated against COVID-19 and previous vaccination against seasonal influenza. Furthermore, we determined the differences in the sources of information, risk perception, and cognitive appraisals of vaccination against COVID-19 between the latent classes. On the basis of the results, we suggest that the various motivations and previous behaviors of vaccination, sources of information about vaccination, risk perception of COVID-19, and cognitive appraisals of vaccination against COVID-19 should be considered in intervention programs aiming to increase the motivation to get vaccinated against COVID-19 and seasonal influenza.

## Figures and Tables

**Figure 1 vaccines-09-01203-f001:**
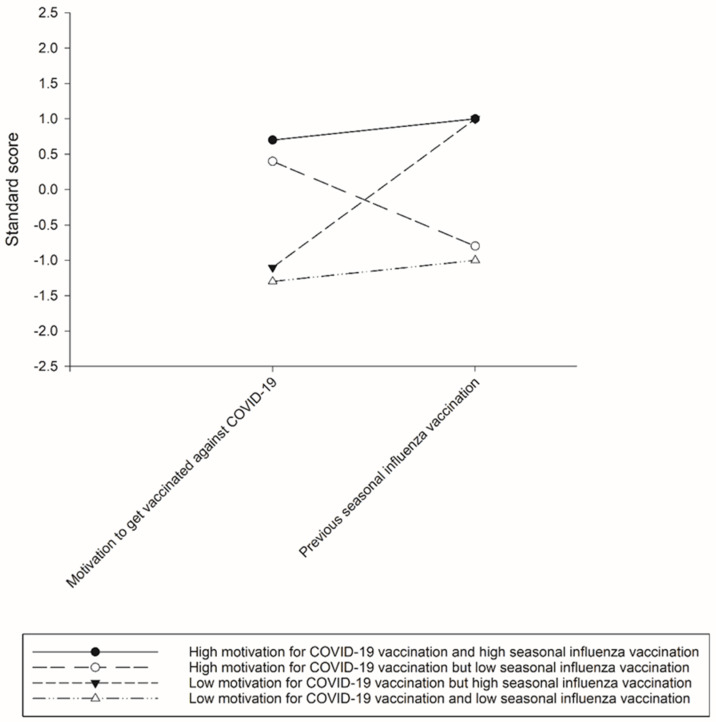
Four classes of participants with various levels of motivation to get COVID-19 vaccination and previous seasonal influenza vaccination.

**Table 1 vaccines-09-01203-t001:** Study questionnaires.

Measures	Items	Response Scale
Motivation to get vaccinated against COVID-19 [41]	Please rate your current willingness to receive a COVID-19 vaccine:	1 (very low) to 10 (very high)
Previous seasonal influenza vaccination	Did you receive seasonal influenza vaccination in the recent years before the COVID-19 pandemic?	1 (never) to 4 (always)
Sources of information concerning COVID-19 vaccination [41]	Do you obtain COVID-19 vaccination information from (1) internet media (e.g., Facebook, Twitter, blogs, and Internet news); (2) traditional media (e.g., newspapers, television, and radio broadcasting); (3) friends; and (4) family members?	0 = no 1 = yes
Perceived risk of COVID-19 [42]	1. If you were to develop flu-like symptoms tomorrow, would you worry?	1 = not at all worried, 2 = worried less than normal, 3 = about the same, 4 = worried more than normal, 5 = extremely worried
	2. In the past week, have you ever worried about catching COVID-19?	1 = no, never think about it, 2 = think about it but it does not worry me, 3 = worried a bit, 4 = worried a lot, 5 = worried about it all the time
	3. Please rate the current level of your worry toward COVID-19:	Scores ranged from 1 to 10 (1 = very mild, 10 = very severe)
	4. How likely do you think that you will contract COVID-19 over the next month?	1 = never, 2 = very unlikely, 3 = unlikely, 4 = evens, 5 = likely, 6 = very likely, 7 = certain
	5. What do you think are your chances of contracting COVID-19 over the next month compared with others outside your family?	1 = not at all, 2 = much less, 3 = less, 4 = evens, 5 = more, 6 = much more, 7 = certain
Drivers of COVID-19 Vaccination Acceptance Scale [43]	1. Vaccination is a very effective way to protect me against COVID-19.	1 = strongly disagree, 2 = disagree, 3 = slightly disagree, 4 = neither disagree nor agree, 5 = slightly agree, 6 = agree, 7 = strongly agree *: reverse-coded
	2. I know very well how vaccination protects me from COVID-19.	
	3. It is important that I get the COVID-19 jab.	
	4. Vaccination greatly reduces my risk of catching COVID-19.	
	5. I understand how the flu jab helps my body fight the COVID-19 virus.	
	6. The COVID-19 jab plays an important role in protecting my life and those of others.	
	7. * I feel under pressure to get the COVID-19 jab.	
	8. The contribution of the COVID-19 jab to my health and well-being is very important.	
	9. I can choose whether to get a COVID-19 jab or not.	
	10. * How the COVID-19 jab works to protect my health is a mystery to me.	
	11. * I get the COVID-19 jab only because I am required to do so.	
	12. Getting the COVID-19 jab has a positive influence on my health.	

COVID-19 = coronavirus disease 2019.

**Table 2 vaccines-09-01203-t002:** Summary of information for selecting the number of latent classes for latent profile analysis.

No. of Classes	AIC	BIC	Entropy	BLRT (*p*-Value)
1	5948.51	5968.33	1	–
2	5243.8	5278.48	0.99	0.01
3	5230.83	5280.36	0.79	0.01
4	5152.79	5217.19	0.83	0.01
5	5122.19	5201.44	0.82	0.01
6	5056.13	5150.25	0.87	0.01

AIC = Akaike information criterion, BIC = Bayesian information criterion, BLRT = bootstrapped likelihood ratio test.

**Table 3 vaccines-09-01203-t003:** Comparison of sociodemographics, sources of information, perceived risks, and cognitive appraisals of vaccination against COVID-19 between various latent classes of motivation to get vaccinated against COVID-19 and previous seasonal influenza vaccination: multinomial logistic regression.

Variable	Both High (*N* = 366)	High COVID-19 but Low Influenza (*N* = 354)	OR 1 ^c^ (95% CI)	Low COVID-19 but High Influenza (*N* = 134)	OR 2 ^c^ (95% CI)	Both Low (*N* = 193)	OR 3 ^c^ (95% CI)
Gender ^a^							
Female	206 (56.3%)	198 (55.9%)	1.00	95 (70.9%)	1.00	118 (61.1%)	1.00
Male	160 (43.7%)	156 (44.1%)	1.01 (0.76–1.36)	39 (29.1%)	0.53 (0.35–0.81) **	75 (38.9%)	0.82 (0.57–1.17)
Age ^a^							
<35	185 (50.5%)	206 (58.2%)	2.62 (1.44–4.78) **	64 (47.8%)	1.15 (0.57–2.33)	87 (45.1%)	0.67 (0.39–1.16)
35–49	141 (38.5%)	131 (37.0%)	2.19 (1.18–4.04) **	58 (43.3%)	1.37 (0.67–2.80)	78 (40.4%)	0.79 (0.45–1.38)
≥50	40 (10.9%)	17 (4.8%)	1.00	12 (9.0%)	1.00	28 (14.5%)	1.00
Education levels ^a^							
High school or below	23 (6.3%)	46 (13.0%)	2.72 (1.53–4.84) **	7 (5.2%)	1.07 (0.42–2.73)	28 (14.5%)	3.28 (1.70–6.31) **
Bachelor’s degree	230 (62.8%)	225 (63.6%)	1.33 (0.95–1.87)	95 (70.9%)	1.46 (0.92–2.31)	123 (63.7%)	1.44 (0.95–2.18)
Master’s degree and above	113 (30.9%)	83 (23.4%)	1.00	32 (23.9%)	1.00	42 (21.8%)	1.00
Health care workers ^a^	169 (46.2%)	23 (6.5%)	0.08 (0.05–0.13) ***	72 (53.7%)	1.35 (0.91–2.01)	15 (7.8%)	0.10 (0.06–0.17) ***
Information from Internet ^b^	309 (84.4%)	282 (79.7%)	0.68 (0.45–1.04)	104 (77.6%)	0.69 (0.42–1.15)	136 (70.5%)	0.43 (0.27–0.67) ***
Information from traditional media ^b^	267 (73.0%)	247 (69.8%)	0.89 (0.63–1.27)	98 (73.1%)	1.04 (0.66–1.64)	126 (65.3%)	0.70 (0.47–1.05)
Information from friend ^b^	128 (35.0%)	116 (32.8%)	1.10 (0.78–1.56)	39 (29.1%)	0.71 (0.46–1.10)	41 (21.2%)	0.57 (0.37–0.89) **
Information from families ^b^	116 (31.7%)	101 (28.5%)	0.74 (0.52–1.05)	44 (32.8%)	1.05 (0.68–1.61)	35 (18.1%)	0.40 (0.25–0.63) ***
Risk perception ^b^	18.3 ± 5.2	17.5 ± 5.5	0.97 (0.95–1.00)	18.2 ± 5.0	1.00 (0.96–1.04)	15.9 ± 5.7	0.92 (0.89–0.95) **
Impact of COVID-19 vaccination ^b^	16.2 ± 2.7	15.7 ± 2.9	0.90 (0.85–0.96) *	12.2 ± 3.4	0.64 (0.59–0.69) ***	12.1 ± 3.5	0.62 (0.57–0.67) ***
Knowledge about COVID-19 vaccination ^b^	15.5 ± 3.6	14.9 ± 3.9	0.93 (0.89–0.97) *	12.2 ± 3.7	0.79 (0.75–0.84) ***	12.4 ± 3.8	0.78 (0.74–0.82) ***
Value of COVID-19 vaccination ^b^	17.4 ± 2.5	16.6 ± 3.0	0.87 (0.82–0.93) ***	13.0 ± 3.6	0.65 (0.60–0.70) ***	12.2 ± 4.0	0.58 (0.53–0.62) ***
Autonomy of COVID-19 vaccination ^b^	18.4 ± 2.8	17.5 ± 2.8	0.89 (0.84–0.94) **	15.0 ± 2.7	0.64 (0.58–0.70) ***	15.8 ± 2.9	0.72 (0.67–0.77) ***

^a^ Unadjusted multinomial logistic regression; ^b^ Multinomial logistic regression with adjustment for sociodemographics; ^c^ High motivations for vaccination against COVID-19 and seasonal influenza as the reference. * *p* < 0.05; ** *p* < 0.01; *** *p* < 0.001.

## Data Availability

The data are available on reasonable request to the corresponding authors.

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
