# Peer review of "Differences in Sources of Information, Risk Perception, and Cognitive Appraisals between People with Various Latent Classes of Motivation to Get Vaccinated against COVID-19 and Previous Seasonal Influenza Vaccination: Facebook Survey Study with Latent Profile Analysis in Taiwan"

_vaccines, 2021, doi:10.3390/vaccines9101203_

Round 1
Reviewer 1 Report
Overall, the study is of potential interest for Vaccines. My main concern is generalizability of the study results to at least the whole Taiwan population. For instance, the median age in Taiwan is about 43 years, while in the study the population 50+ years accounted for only 10%. Older adults are the main target for both COVID-19 and flu vaccines but these were underrepresented likely due to the recruitment on Facebook. Social media users are also more exposed to incorrect information on vaccines. This means that the identified clusters are likely biased from the point of view of numerosity. Moreover, the factors associated with the “membership” to a given cluster may be biased. I therefore believe that the study results may be generalizable to Taiwanese Facebook users and this fact should be mentioned in the manuscript title.
The Introduction is a bit long with an impressive number of citations (N=52). I recommend to focus on what is already known on the topic, the study hypothesis and objectives. Moreover, several studies on COVID-19 vaccine hesitancy/acceptance conducted in Taiwan are available (see for example https://pubmed.ncbi.nlm.nih.gov/?term=%22taiwan%22+and+%22covid%22+and+%22hesitancy%22&sort=date) but not presented in the Introduction.
The questionnaire item “Sources of information concerning COVID-19 vaccination” includes only 4 response options (new and traditional media, family and friends). I wonder why the role of physicians, pharmacists, public health authorities was not investigated? These are usually the most trusted sources of health-related information including COVID-19 vaccines.
The selection of the number of latent classes is unclear. In the methods you stated that this number was selected on the basis of AIC/BIC minimization criterion and higher entropy. From Table 2 it seems that a 6-class model performs better than the 4-class model selected.
Author Response
We appreciated your valuable comments. As discussed below, we have revised our manuscript based on your suggestions. Please let us know if we need to provide anything else regarding this revision.
Comment 1
My main concern is generalizability of the study results to at least the whole Taiwan population. For instance, the median age in Taiwan is about 43 years, while in the study the population 50+ years accounted for only 10%. Older adults are the main target for both COVID-19 and flu vaccines but these were underrepresented likely due to the recruitment on Facebook. Social media users are also more exposed to incorrect information on vaccines. This means that the identified clusters are likely biased from the point of view of numerosity. Moreover, the factors associated with the “membership” to a given cluster may be biased. I therefore believe that the study results may be generalizable to Taiwanese Facebook users and this fact should be mentioned in the manuscript title.
Response
Thank you for your reminding. We added “Facebook Survey Study with Latent Profile Analysis in Taiwan” into the manuscript title. We also added more discussion regarding to inviting participants on Facebook in the limitation as below. Please refer to line 5-6.
“However, Facebook has a younger audience than the general population [56,57]. Participants who were 50 years old or older accounted for only 9.2% of all participants in the present study. Older adults are the main target for both COVID-19 and seasonal influenza vaccines; they might be underrepresented due to the recruitment on Facebook. Moreover, social media users are also more exposed to incorrect information on vaccines [25]. Therefore, the results in our study may not be generalizable to the general population.”
Comment 2
The Introduction is a bit long with an impressive number of citations (N=52). I recommend to focus on what is already known on the topic, the study hypothesis and objectives. Moreover, several studies on COVID-19 vaccine hesitancy/acceptance conducted in Taiwan are available (see for example https://pubmed.ncbi.nlm.nih.gov/?term=%22taiwan%22+and+%22covid%22+and+%22hesitancy%22&sort=date) but not presented in the Introduction.
Response
Thank you for your suggestion. We shortened the content of Introduction section and focused on the topic, hypothesis and objectives of this study. The number of citations was reduced from 52 to 38. We also added the results of two studies on COVID-19 vaccine hesitancy/acceptance conducted in Taiwan into the revised manuscript (references 20 and 30).
“By using the Protection Motivation Theory [19], perceived knowledge was significantly associated with intention to get vaccinated against COVID-19 [20].” Please refer to line 59-61.
“A previous study in Taiwan revealed that the risk perception of influenza infection was positively associated with intention to receive a vaccine against influenza [30].” Please refer to line 77-79.
Comment 3
The questionnaire item “Sources of information concerning COVID-19 vaccination” includes only 4 response options (new and traditional media, family and friends). I wonder why the role of physicians, pharmacists, public health authorities was not investigated? These are usually the most trusted sources of health-related information including COVID-19 vaccines.
Response
Thank you for your comment. We totally agree that medical professionals are the most reliable source of health-related information regarding COVID-19. However, 26.6% of participants in this study were health care workers; therefore, we selected the variable “health care workers” instead of “get information of COVID-19 vaccines from medical professionals” into analysis. We added the explanation into the limitation section of the revised manuscript. Please refer to line 351-355.
“Third, research found that medical professionals are the most reliable source of health-related information regarding COVID-19 [58]. However, given that 26.6% of participants in this study were health care workers, we did not analyze the role of getting information of COVID-19 vaccines from medical professionals in the latent classes of motivation to get vaccinated against COVID-19 and previous seasonal influenza vaccination.”
Comment 4
The selection of the number of latent classes is unclear. In the methods you stated that this number was selected on the basis of AIC/BIC minimization criterion and higher entropy. From Table 2 it seems that a 6-class model performs better than the 4-class model selected.
Response
Thank you for your comment. We did not select the 6-class model because it is not rational to select an LPA model with a great number of classes, especially for a model that has only two variables (i.e., motivation to get vaccinated against COVID-19 and previous seasonal influenza vaccination). We have added more information in the Method and highlighted our explanation for the selection of the number of latent classes in Results.
Method
To determine how many latent classes obtained from LPA, we used the basic model according to four model fit indices: Akaike information criterion (AIC), Bayesian information criterion (BIC), entropy, and bootstrapped likelihood ratio test (BLRT). The model with lower AIC and BIC has a better fit than those with higher AIC and BIC. An entropy value approaching 1 indicates a clear separation of classes [47], and entropy >0.80 indicates that the latent classes are highly discriminating [48]. For BLRT, p < 0.05 indicates that the k class model is superior to the k − 1 class model (k represents the number of classes). “If there was no best fit model, we would select the number of latent classes based on the parsimony principle and the literature.” Please refer to line 187-189.
Results
“Although the model with two latent classes had an inflection point of BIC, BIC and AIC continuously decreased when the number of classes increased; statistical significance was based on BLRT. It is not rational to select an LPA model with a great number of classes, especially for a model that has only two variables (i.e., motivation to get vaccinated against COVID-19 and previous seasonal influenza vaccination). Thus, we chose the best model based on the parsimony principle and the literature [49]; the number of classes in LPA usually ranges from two to four.” Please refer to line 202-208.
Reference 49: Nagin, D. Group-Based Modeling of Development. Cambridge, MA, USA: Harvard University Press; 2005, p 74.
Reviewer 2 Report
Thank you for submitting an interesting article!
Please review the below for suggestions:
1) The MN is a bit worthy, it can be shortened to have only the vital information for the readers to grasp the purpose of the article.
2) Per 1.4, I would change distinct classes -> distinct latent classes for possible ambiguity
3) For recruiting participants via Facebook advertisement, could you please capture the ads and submit them as a supplement? (Please review the attached file) In addition, a detailed explanation for recruiting participants via Facebook is a must for ensuring transparency. Please re-consider adding more to the 2.1 Participants section.
4) Table 1 can be moved to the next page, and could you make spaced between the questions?
5) Did you perhaps use the validated questionnaire? If so, please cite them in the reference section.
6) Regarding 3.1 Results of LPA -- "Thus, we chose the best model based on the parsimony principle and the literature" -- I believe this sentence needs a reference.
7) As per Table 3, did you mention why you have chosen Female, Age less than 50, Master's degree and above for comparison when performing MLR?
8) With regards to 4.3 Implication, the first sentence mentioned that the present study is one of the first studies. If there are other studies, I would recommend citing them as references.
Thank you!

Author Response
Comment
1) The MN is a bit worthy, it can be shortened to have only the vital information for the readers to grasp the purpose of the article.
Response
Thank you for your suggestion. We shortened the content of Introduction section and focused on the topic, hypothesis and objectives of this study. The number of words in Introduction reduced from 914 to 686. Please refer to line 46-107.
Comment
2) Per 1.4, I would change distinct classes -> distinct latent classes for possible ambiguity
Response
We changed it into “distinct latent classes”. Please refer to line 95.
Comment
3) For recruiting participants via Facebook advertisement, could you please capture the ads and submit them as a supplement? (Please review the attached file) In addition, a detailed explanation for recruiting participants via Facebook is a must for ensuring transparency. Please re-consider adding more to the 2.1 Participants section.
Response
We submitted the Facebook advertisement for recruiting the participants as the supplement. We also added the contents of 2.1. Participants to explain the method of recruiting participants via Facebook as below. Please refer to line 111-122.
“In brief, 1047 participants were recruited by using a Facebook advertisement on 15 October 2020 and 21 December 2020. Facebook (Facebook Inc., Menlo Park, CA, USA) users were eligible for this study if they were ≥20 years old and living in Taiwan. The Facebook advertisement included a headline, main text, pop-up banner, and weblink to the research questionnaire website. We designed the advertisement to appear in the “News Feed” of Facebook, which is a streaming list of updates from the user’s connections (e.g., friends) and advertisers. We focused solely on News Feed advertisements (Facebook), as opposed to other Facebook advertising locations (e.g., the right column), because News Feed advertisements are more effective in terms of recruitment metrics for research studies. We targeted the advertisement to Facebook users by location (Taiwan) and by language (Mandarin Chinese), such that a given advertisement appeared on a user’s news feed as determined by a Facebook algorithm.”
Comment
4) Table 1 can be moved to the next page, and could you make spaced between the questions?
Response
In the revised manuscript we put the whole Table 1 into one page without segmentation. Please refer to page 4. We also added lines into the table to make the questions easily read.
Comment
5) Did you perhaps use the validated questionnaire? If so, please cite them in the reference section.
Response
Thank you for your reminding. We cited them (references 41-43) in the reverence section.
Comment
6) Regarding 3.1 Results of LPA -- "Thus, we chose the best model based on the parsimony principle and the literature" -- I believe this sentence needs a reference.
Response
This statement is based on Nagin (2005), which is the following paragraph. We added this reference in the revised manuscript. Please refer to line 208.
Reference 49: Nagin, D. Group-Based Modeling of Development. Cambridge, MA, USA: Harvard University Press; 2005, p 74.
Comment
7) As per Table 3, did you mention why you have chosen Female, Age less than 50, Master's degree and above for comparison when performing MLR?
Response
We added the description as below to illustrate why we adjusted the effects of gender, age and education level when examining the differences in sources of information, risk perception, and cognitive appraisals between various latent classes.
“Research has found that being female [9], younger age [37,38], and lower educational attainment [37] were significantly associated with lower motivation to receive COVID-19 vaccination; therefore, we adjusted the effects of gender, age and education level when examining the differences in sources of information, risk perception, and cognitive appraisals between various latent classes.” Please refer to line 103-107.
“Firstly, differences in sex, age, educational level, and occupation between various latent classes were examined using unadjusted multinomial logistic regression.” Please refer to line 246-248.
“Differences in information sources between various latent classes were further examined using adjusted multinomial logistic regression. Because of significant differences in sex, age, educational level, and occupation between latent classes, these sociodemographics were adjusted in multinomial logistic regression.” Please refer to line 259-262.
Comment
8) With regards to 4.3 Implication, the first sentence mentioned that the present study is one of the first studies. If there are other studies, I would recommend citing them as references.
Response
To our best knowledge, there was no other studies classifying individuals with various levels of motivation to get vaccinated against COVID-19 and previous seasonal influenza vaccination. We revised this sentence as below.
“The present study is the first one to…” Please refer to line 325.
Round 2
Reviewer 1 Report
All my comments were addressed. I have no further comments. The paper, however, has to be reviewed by a Native English speaker.
Author Response
Thank you for your support. We have sent the revised manuscript to another native English editor. therefore, this manuscript was edited twice. We hope the writing quality has been improved.
Reviewer 2 Report
- Could you please re-check if you have uploaded the supplements? I am not able to download to view the Facebook advertisement.
Author Response
It seems to have some problems in showing the supplementary file. Therefore, in addition to re-upload the supplementary file containing the Facebook advertisement, we added it at the end of the revised manuscript as Appendix. Please let us know if anything else we should provide.